# Hospital-treated infections in early- and mid-life and risk of Alzheimer's disease, Parkinson's disease, and amyotrophic lateral sclerosis: A nationwide nested case-control study in Sweden

Jiangwei Sun[1]*, Jonas F. Ludvigsson[2,3,4], Caroline Ingre[5], Fredrik Piehl[5], Karin Wirdefeldt[2,5], Ulrika Zagai[2], Weimin Ye[2], Fang Fang[1]

1 Institute of Environmental Medicine, Karolinska Institutet, Stockholm, Sweden, 2 Department of Medical Epidemiology and Biostatistics, Karolinska Institutet, Stockholm, Sweden, 3 Department of Pediatrics, Örebro University Hospital, Örebro, Sweden, 4 Department of Medicine, Columbia University College of Physicians and Surgeons, New York, New York, United States of America, 5 Department of Clinical Neuroscience, Karolinska Institutet, Stockholm, Sweden

* jiangwei.sun@ki.se

**Data Availability Statement:** The data set cannot be shared directly under current legislation for data

## Abstract

### Background

Experimental observations have suggested a role of infection in the etiology of neurodegenerative disease. In human studies, however, it is difficult to disentangle whether infection is a risk factor or rather a comorbidity or secondary event of neurodegenerative disease. To this end, we examined the risk of 3 most common neurodegenerative diseases in relation to previous inpatient or outpatient episodes of hospital-treated infections.

### Methods and findings

We performed a nested case-control study based on several national registers in Sweden. Cases were individuals newly diagnosed with Alzheimer's disease (AD), Parkinson's disease (PD), or amyotrophic lateral sclerosis (ALS) during 1970 to 2016 in Sweden, identified from the National Patient Register. For each case, 5 controls individually matched to the case on sex and year of birth were randomly selected from the general population. Conditional logistic regression was used to estimate odds ratios (ORs) and 95% confidence intervals (CIs) with adjustment for potential confounders, including sex, year of birth, area of residence, educational attainment, family history of neurodegenerative disease, and Charlson comorbidity index. Infections experienced within 5 years before diagnosis of neurodegenerative disease were excluded to reduce the influence of surveillance bias and reverse causation. The analysis included 291,941 AD cases (median age at diagnosis: 76.2 years; male: 46.6%), 103,919 PD cases (74.3; 55.1%), and 10,161 ALS cases (69.3; 56.8%). A hospital-treated infection 5 or more years earlier was associated with an increased risk of AD (OR = 1.16, 95% CI: 1.15 to 1.18, P < 0.001) and PD (OR = 1.04, 95% CI: 1.02 to 1.06, P

protection and must be requested directly from the respective registry holders, Statistics Sweden (information@scb.se) and the Swedish National Board of Health and Welfare (registerservice@socialstyrelsen.se), after approval by the Swedish Ethical Review Authority.

**Funding:** This study was supported by the Swedish Research Council (grants No: 2019-01088 (FF), 340-2013-5867 (FF), and 2017-02175 (KW)), the Joint Program on Neurodegenerative Diseases (JPND, grant number: 2021-00696 (FF)), and the Chinese Scholarship Council (JS). The funders had no role in study design, data collection and analysis, decision to publish, or preparation of the manuscript.

**Competing interests:** I have read the journal's policy and the authors of this manuscript have the following competing interests: JL coordinates a study (independent of the present study) on behalf of the Swedish IBD quality register (SWIBREG). That study has received funding from Janssen Corporation. Other authors declared no competing interests.

**Abbreviations:** AD, Alzheimer's disease; ALS, amyotrophic lateral sclerosis; CI, confidence interval; CNS, central nervous system; OR, odds ratio; PD, Parkinson's disease.

< 0.001). Similar results were observed for bacterial, viral, and other infections and among different sites of infection including gastrointestinal and genitourinary infections. Multiple infections before age 40 conveyed the greatest risk of AD (OR = 2.62, 95% CI: 2.52 to 2.72, $P$ < 0.001) and PD (OR = 1.41, 95% CI: 1.29 to 1.53, $P$ < 0.001). The associations were primarily due to AD and PD diagnosed before 60 years (OR = 1.93, 95% CI: 1.89 to 1.98 for AD, $P$ < 0.001; OR = 1.29, 95% CI: 1.22 to 1.36 for PD, $P$ < 0.001), whereas no association was found for those diagnosed at 60 years or older (OR = 1.00, 95% CI: 0.98 to 1.01 for AD, $P$ = 0.508; OR = 1.01, 95% CI: 0.99 to 1.03 for PD, $P$ = 0.382). No association was observed for ALS (OR = 0.97, 95% CI: 0.92 to 1.03, $P$ = 0.384), regardless of age at diagnosis. Excluding infections experienced within 10 years before diagnosis of neurodegenerative disease confirmed these findings. Study limitations include the potential misclassification of hospital-treated infections and neurodegenerative diseases due to incomplete coverage of the National Patient Register, as well as the residual confounding from unmeasured risk or protective factors for neurodegenerative diseases.

## Conclusions

Hospital-treated infections, especially in early- and mid-life, were associated with an increased risk of AD and PD, primarily among AD and PD cases diagnosed before 60 years. These findings suggest that infectious events may be a trigger or amplifier of a preexisting disease process, leading to clinical onset of neurodegenerative disease at a relatively early age. However, due to the observational nature of the study, these results do not formally prove a causal link.

## Author summary

### Why was this study done?

- Experimental studies suggest that infection plays a role in neurodegenerative disease development. Supporting evidence in humans is, however, scarce.

- Due to the long preclinical stage of neurodegenerative diseases, it is still unclear whether infection constitutes a risk factor or is merely a comorbidity or secondary event.

- No study has explored the association of infections treated in specialized care (i.e., hospital in- and outpatient care) with the subsequent risk of common neurodegenerative diseases (i.e., Alzheimer's disease (AD), Parkinson's disease (PD), and amyotrophic lateral sclerosis (ALS)) in a single population.

### What did the researchers find?

- Infections treated in specialized care were associated with an increased subsequent risk of AD and PD—primarily AD and PD diagnosed before 60 years, but not ALS. These positive associations remained after excluding infections experienced within 10 years before diagnosis of AD or PD.

- Increased risks of AD and PD were observed for bacterial, viral, and other infections, and across different sites of infection, including gastrointestinal and genitourinary infections.

- Individuals with repeated infections in early- and mid-life had the greatest risk increment of AD and PD.

### What do these findings mean?

- The underlying mechanisms for the link between infections and neurodegenerative disease may not be specific to certain pathogens or affected organs but possibly occur at the systemic level.

- Infectious events may be a trigger or amplifier of a preexisting disease process, leading to clinical onset of neurodegenerative disease at a relatively early age among individuals with disease predisposition.

## Introduction

Neurodegenerative diseases, including Alzheimer's disease (AD), Parkinson's disease (PD), and amyotrophic lateral sclerosis (ALS), are characterized by progressive loss of neurons in the nervous systems [1]. Although the incidence and prevalence varied with age, sex, and geography, the global burden of neurodegenerative diseases more than doubled during 1990 to 2016 [2–4]. Both genetic and nongenetic factors contribute to its development [5], but only a small proportion of patients with neurodegenerative diseases are driven by genetic causes [6,7].

A potential infectious etiology has been hypothesized for neurodegenerative diseases, as findings in animal studies have demonstrated that infectious processes might impact pathogenesis, phenotype, and progression of neurodegenerative disease [1,7–9]. Proposed underlying mechanisms include modulating misfolding and aggregation of pathological proteins, neuroinflammation, and infiltration of peripheral immune cells into the central nervous system (CNS) [1,9,10]. The extrapolation of such findings to a human context is, however, not straightforward. Previous studies have mostly examined the role of specific pathogens on a specific neurodegenerative disease, e.g., herpesvirus for AD [11], and influenza [12], hepatitis C virus [13], and *Helicobacter pylori* [14] for PD, with inconclusive results [7–9,11,15,16]. Although several studies have also assessed associations between infectious diseases and risk of dementia [17–19] and AD [20], influence of potential surveillance bias (greater-than-expected surveillance of disease after infections) and reverse causation (due to, for example, diagnostic delay of neurodegenerative diseases) on the associations was not always fully addressed. Therefore, whether infection is indeed a risk factor rather a comorbidity or secondary event of neurodegenerative disease remains unknown. In contrast to AD and PD, the potential link between infection and ALS has been less explored [8,21].

Weak evidence from human studies may be attributed to multiple factors. First, only few studies applied a prospective study design, making it difficult to differentiate causality from epiphenomena. Moreover, given that studies addressed different infections and neurodegenerative diseases, it is challenging to disentangle methodological drawbacks from real biological

difference. Therefore, a comprehensive examination of different infections and different neurodegenerative diseases in a single study population might be of importance, after careful consideration of surveillance bias and reverse causation. Finally, a life course approach may also help to assess potential high-risk time window and critical periods for intervention [15].

To this end, using Swedish national healthcare registers, we performed a nationwide nested case-control study to examine the associations of hospital-treated infections, namely infections requiring inpatient or outpatient care, with the risk of AD, PD, and ALS. Our secondary aim was to explore whether the associations varied by type, site, age, and frequency of infection.

## Methods

### Study design

All individuals born after 1900 in Sweden whose parents were also born in Sweden were eligible for this study ($N$ = 12,275,551). We followed these individuals from 1970 until a diagnosis of neurodegenerative disease, emigration, death, or December 31, 2016, whichever occurred first, through cross-linkages to the National Patient Register and the Causes of Death Register, using the individually unique Swedish personal identity number. The National Patient Register was established in 1964 and started to include all inpatient care since 1987; it also includes over 80% of outpatient care since 2001 [22]. We used this register to identify new diagnoses of neurodegenerative diseases via the Swedish revisions of the *International Classification of Disease* (ICD) codes, considering both primary and secondary diagnoses (S1 Table). Individuals with a diagnosis of neurodegenerative diseases before 1970 were excluded from the analyses, and individuals with multiple neurodegenerative diseases during follow-up contributed to the analyses of different diseases. The register-based definitions of AD, PD, and ALS have been validated against gold-standard clinical workup, showing a high specificity but a varying sensitivity and positive predictive value for AD (99.7%, 32.5%, and 57%) [23], PD (>98%, 72%, and 71%) [24], and ALS (all >90%) [25]. The relatively low sensitivity of AD diagnosis is likely attributable to misdiagnosis of AD as other dementias [23]. Date of diagnosis was defined as the date of first hospital visit concerning the disease as either the primary or a secondary diagnosis.

A nested case-control study within the above study base was then conducted to assess the associations of hospital-treated infections with the risk of AD, PD, and ALS. Five controls per case, individually matched by sex and year of birth, were randomly selected from the study base using the method of incidence density sampling [26]. Date of diagnosis and date of selection were used as the index date for cases and controls, respectively. Controls should be alive and free of the specific neurodegenerative disease of their matched case when selected. The prespecified analysis plan is presented in S1 Text.

### Hospital-treated infections

Hospital inpatient and outpatient visits with a diagnosis of infection before index date were identified from the National Patient Register, using ICD codes shown in S2 Table [27]. We first studied any infection as a binary variable and then studied infections by type (bacterial, viral, or other infection), site (CNS, gastrointestinal, genitourinary, respiratory, or skin infection), age (<40, 40 to 59.9, or ≥60 years), and frequency (0, 1, or ≥2 events).

### Covariates

We identified information on area of residence (3 groups: Northern, Central, and Southern Sweden) from the Total Population Register (information available from 1947 onward) and

educational attainment (4 groups: 0 to 9 years, 10 to 12 years, ≥13 years, and "missing") from the Swedish Longitudinal Integrated Database for Health Insurance and Labour Market Studies (information available from 1990 onward) [28]. Family history of neurodegenerative disease (yes/no) was defined as a diagnosis of the disease among first-degree relatives (i.e., biological parents and full siblings) (information available from 1964 onward). We identified parents and full siblings from the Swedish Multi-Generation Register [29]. Charlson comorbidity index (3 groups: 0, 1, and ≥2), as a proxy of general health status, was calculated using data from the National Patient Register (information available from 1964 onward) [30]. All covariates were measured on the index date of the cases and controls.

## Statistical analyses

We performed separate analyses for AD, PD, and ALS. If 1 individual was diagnosed with more than one of the studied neurodegenerative diseases, for example, a diagnosis of PD first and a diagnosis of AD later, this individual would contribute to the analysis of PD and AD with the respective diagnosis date. We first described the percentage of individuals with a history of infections among cases and controls during 20 years before index date, and then applied conditional logistic regression to estimate odds ratio (OR) and 95% confidence interval (CI), as an estimate of the association between infections and risk of neurodegenerative disease. As OR obtained from a nested case-control study within a well-defined underlying cohort and with controls selected using the method of incidence density sampling is mathematically equal to relative risk estimate of the underlying cohort study [31]. OR in the present study can be interpreted as incidence rate ratio [32]. In addition to conditioning on matching variables (sex and year of birth), we adjusted for area of residence, educational attainment, family history of neurodegenerative disease, and Charlson comorbidity index (i.e., history of comorbidity) in the analysis. Due to the concern of surveillance bias and reverse causation [23,24,33], we used a lag time of 5 years, namely all infections experienced during 5 years before the index date were excluded from the analysis.

The analyses were first performed for any infection and then by type, site, age, and frequency of infection. To examine the potential dose–response relationship within specific age groups, we also analyzed frequency of infection by age at infection. We stratified the analysis by sex (male or female), age at index date (<60 years or ≥60 years), calendar period (1970–1986, 1987–2000, and 2001–2016), and year of birth (1900 to 1919, 1920 to 1939, 1940 to 1959, or ≥1960) to assess whether the associations would differ by sex, age, calendar period, and birth cohort. As previous studies have used age 65 to define early- and late-onset AD [34], we additionally performed a stratified analysis of AD by age 65 at index date. The P value for interaction was calculated using the Wald test for the product terms between infection and a specific covariate. Because the positive associations were mainly noted for those diagnosed before 60 years, and the etiologies for AD and PD diagnosed at relatively young age are potentially different from those diagnosed at later age, we repeated all analyses for AD, PD, and ALS diagnosed before or after 60 years, separately.

We also performed a series of secondary analyses to assess the robustness of the findings. (1) To assess whether the associations could be modified by family history of neurodegenerative disease, in a sensitivity analysis, we restricted the analysis to those without a family history of the disease. (2) To minimize potential misclassification of outcomes, we, in another sensitivity analysis, restricted the definition of outcomes to those with at least 2 hospital visits concerning the same disease. (3) Because information on education was available only from 1990 onward, there was a high degree of missingness for this covariate. To assess to what extent this could influence the results, we repeated the main analysis without adjustment for education and additionally compared the results between 2 models with or without adjustment for

education among individuals with data available on education. (4) We additionally excluded individuals with more than 1 neurodegenerative disease from the analysis (2.64% with AD and PD, 0.05% with AD and ALS, 0.04% with PD and ALS, and <0.01% with all 3 diseases). (5) Finally, to assess the impact of choice of lag time, we performed another sensitivity analysis by excluding infections experienced during 10 years before the index date.

Data analyses were performed using SAS version 9.4 (SAS Institute, Cary, NC) and R version 3.6.0. A two-sided $P \leq 0.05$ was considered statistically significant. This study is reported according to the Strengthening the Reporting of Observational Studies in Epidemiology (STROBE) guideline (S1 STROBE Checklist).

### Ethics consideration

The study was approved by the Regional Ethical Review Board in Stockholm (2012/1814-31/4). According to the current Swedish regulation, informed consent is not required when register data are used for the purpose of research.

### Results

We enrolled 291,941 AD cases (median age at diagnosis: 76.2 years; male: 46.6%), 103,919 PD cases (74.3; 55.1%), and 10,161 ALS cases (69.3; 56.8%), together with their matched controls in the study (Table 1). Compared with controls, individuals with neurodegenerative diseases

**Table 1. Characteristics of patients with neurodegenerative diseases and their matched controls at the index date.**

| | AD | | PD | | ALS | |
|---|---|---|---|---|---|---|
| | **Case** | **Control** | **Case** | **Control** | **Case** | **Control** |
| N | 291,941 | 1,459,705 | 103,919 | 519,595 | 10,161 | 50,805 |
| Age at the index date, years | | | | | | |
| Mean ± SD | 67.7 ± 21.2 | 67.7 ± 21.2 | 71.5 ± 13.8 | 71.5 ± 13.8 | 67.7 ± 12.1 | 67.7 ± 12.1 |
| Median (Q1-Q3) | 76.2 (58.0–82.6) | 76.2 (58.0–82.6) | 74.3 (67.1–79.9) | 74.3 (67.1–79.9) | 69.3 (61.3–76.1) | 69.3 (61.4–76.1) |
| <60 years, n (%) | 76,023 (26.0) | 380,101 (26.0) | 13,319 (12.8) | 66,661 (12.8) | 2,245 (22.1) | 11,216 (22.1) |
| ≥60 years, n (%) | 215,918 (74.0) | 1,079,604 (74.0) | 90,600 (87.2) | 452,934 (87.2) | 7,916 (77.9) | 39,589 (77.9) |
| Sex, n (%) | | | | | | |
| Male | 135,998 (46.6) | 679,990 (46.6) | 57,281 (55.1) | 286,405 (55.1) | 5,773 (56.8) | 28,865 (56.8) |
| Female | 155,943 (53.4) | 779,715 (53.4) | 46,638 (44.9) | 233,190 (44.9) | 4,388 (43.2) | 21,940 (43.2) |
| Area of residence, n (%) | | | | | | |
| Northern Sweden | 65,230 (22.3) | 353,015 (24.2) | 25,162 (24.2) | 125,333 (24.1) | 2,316 (22.8) | 12,224 (24.1) |
| Central Sweden | 162,484 (55.7) | 775,595 (53.1) | 55,654 (53.6) | 275,688 (53.1) | 5,589 (55.0) | 26,994 (53.1) |
| Southern Sweden | 64,227 (22.0) | 331,095 (22.7) | 23,103 (22.2) | 118,574 (22.8) | 2,256 (22.2) | 11,587 (22.8) |
| Educational attainment, n (%) | | | | | | |
| 0–9 years | 77,523 (26.6) | 388,391 (26.6) | 29,081 (28.0) | 153,830 (29.6) | 2,922 (28.8) | 15,586 (30.7) |
| 10–12 years | 45,667 (15.6) | 219,121 (15.0) | 18,903 (18.2) | 92,500 (17.8) | 2,725 (26.8) | 12,694 (25.0) |
| ≥13 years | 16,711 (5.7) | 91,564 (6.3) | 10,076 (9.7) | 42,998 (8.3) | 1,393 (13.7) | 6,751 (13.3) |
| Missing | 152,040 (52.1) | 760,629 (52.1) | 45,859 (44.1) | 230,267 (44.3) | 3,121 (30.7) | 15,774 (31.1) |
| Family history of the disease, n (%) | 18,345 (6.3) | 54,522 (3.7) | 2,377 (2.3) | 7,088 (1.4) | 181 (1.8) | 126 (0.3) |
| Charlson comorbidity index, n (%) | | | | | | |
| 0 | 251,119 (86.0) | 1,220,639 (83.6) | 87,590 (84.3) | 436,444 (84.0) | 8,834 (86.9) | 43,972 (86.6) |
| 1 | 37,134 (12.7) | 215,219 (14.7) | 14,604 (14.1) | 74,907 (14.4) | 1,197 (11.8) | 6,171 (12.2) |
| ≥2 | 3,688 (1.3) | 23,847 (1.6) | 1,725 (1.7) | 8,244 (1.6) | 130 (1.3) | 662 (1.3) |

AD, Alzheimer's disease; ALS, amyotrophic lateral sclerosis; PD, Parkinson's disease; SD, standard deviation.

Index date: date of diagnosis for cases and date of selection for controls.

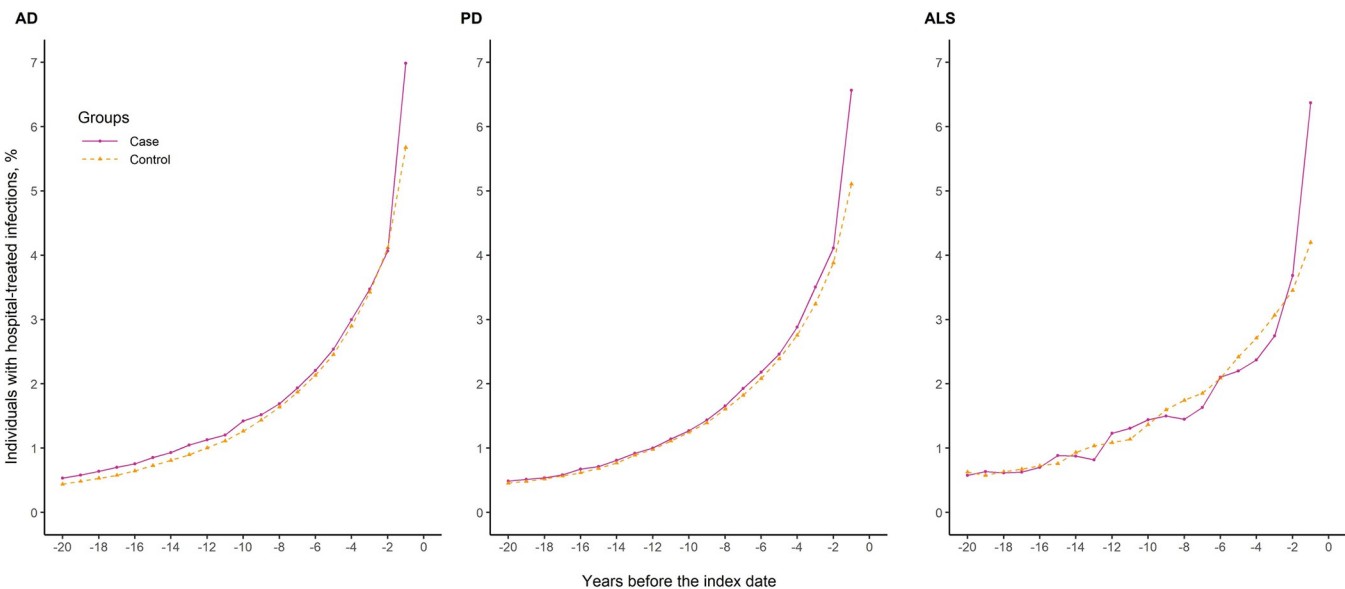

**Fig 1. Percentage of individuals with hospital-treated infections among patients with neurodegenerative diseases and their matched controls during the 20 years before the index date.** AD, Alzheimer's disease; ALS, amyotrophic lateral sclerosis; PD, Parkinson's disease.

had a higher percentage of family history of neurodegenerative disease but lower Charlson comorbidity index score.

Compared with controls, there was a slightly higher percentage of individuals with hospital-treated infections among patients with AD or PD, but not ALS, during the 20 years before the index date (Fig 1). The percentage increased rapidly during the year before diagnosis of neurodegenerative disease.

### Primary analyses

After excluding infections diagnosed during 5 years before the index date, an event of hospital-treated infection was associated with a higher risk of AD (OR = 1.16, 95% CI: 1.15 to 1.18, $P < 0.001$) and PD (OR = 1.04, 95% CI: 1.02 to 1.06, $P < 0.001$), but not ALS (OR = 0.97, 95% CI: 0.92 to 1.03, $P = 0.384$) (Table 2). Positive associations for AD and PD were similarly observed for bacterial, viral, and other infections, as well as for gastrointestinal and genitourinary infections (Table 2). For ALS, we did not observe any association.

Magnitude of the associations of any hospital-treated infections with the risk for AD and PD decreased with increasing age at infection, with the strongest associations noted for infection at age 40 or below (OR = 1.86, 95% CI: 1.82 to 1.90 for AD, $P < 0.001$; OR = 1.20, 95% CI: 1.15 to 1.25 for PD, $P < 0.001$) (S3 Table). Dose–response relationships were observed for frequency of infections before 40 years (all $P_{for\ trend} < 0.001$). For example, individuals with $\geq 2$ events of infections before 40 years had the highest risk of AD (OR = 2.62, 95% CI: 2.52 to 2.72, $P < 0.001$) and PD (OR = 1.41, 95% CI: 1.29 to 1.53, $P < 0.001$) (S3 Table).

In the stratified analyses of any hospital-treated infection, a stronger association was observed among male (OR = 1.20, 95% CI: 1.18 to 1.22, $P < 0.001$) than female (OR = 1.13, 95% CI: 1.12 to 1.15, $P < 0.001$) for AD, but among female (OR = 1.07, 95% CI: 1.05 to 1.11, $P < 0.001$) than male (OR = 1.01, 95% CI: 0.98 to 1.03, $P = 0.536$) for PD (both $P_{for\ interaction} <$ 0.001, S4 Table). The associations were primarily limited to individuals diagnosed before 60 years (OR = 1.93, 95% CI: 1.89 to 1.98 for AD, $P < 0.001$; OR = 1.29, 95% CI: 1.22 to 1.36 for

**Table 2. Association between hospital-treated infection and risk of neurodegenerative disease.**

| Group | AD | | | | PD | | | | ALS | | | |
|---|---|---|---|---|---|---|---|---|---|---|---|---|
| | Infection (case/control) | No infection (case/control) | OR (95% CI) | P | Infection (case/control) | No infection (case/control) | OR (95% CI) | P | Infection (case/control) | No infection (case/control) | OR (95% CI) | P |
| Any infection | 49,789/225,776 | 242,152/1,233,929 | 1.16 (1.15–1.18) | <0.001 | 16,818/81,891 | 87,101/437,704 | 1.04 (1.02–1.06) | <0.001 | 1,793/9,184 | 8,368/41,621 | 0.97 (0.92–1.03) | 0.384 |
| Infection type | | | | | | | | | | | | |
| Bacterial | 27,182/120,724 | 264,759/1,338,981 | 1.16 (1.15–1.18) | <0.001 | 9,113/44,038 | 94,806/475,557 | 1.04 (1.02–1.07) | 0.001 | 983/4949 | 9,178/45,856 | 1.00 (0.93–1.08) | 0.970 |
| Viral | 14,136/64,001 | 277,805/1,395,704 | 1.12 (1.10–1.15) | <0.001 | 4,860/22,910 | 99,059/496,685 | 1.07 (1.03–1.10) | <0.001 | 549/2,618 | 9,612/48,187 | 1.05 (0.95–1.16) | 0.324 |
| Others | 5,336/23,658 | 286,605/1,436,047 | 1.15 (1.11–1.18) | <0.001 | 1,886/8,826 | 102,033/510,769 | 1.07 (1.02–1.13) | 0.006 | 252/1,165 | 9,909/49,640 | 1.09 (0.95–1.26) | 0.219 |
| Infection site | | | | | | | | | | | | |
| CNS | 1,865/8,354 | 290,076/1,451,351 | 1.13 (1.07–1.19) | <0.001 | 672/3,133 | 103,247/516,462 | 1.07 (0.99–1.17) | 0.097 | 78/436 | 10,083/50,369 | 0.89 (0.69–1.13) | 0.335 |
| Gastrointestinal | 9,038/39,856 | 282,903/1,419,849 | 1.23 (1.20–1.26) | <0.001 | 3,041/14,386 | 100,878/505,209 | 1.07 (1.02–1.11) | 0.002 | 326/1,776 | 9,835/49,029 | 0.92 (0.81–1.04) | 0.188 |
| Respiratory | 17,649/85,976 | 274,292/1,373,729 | 1.04 (1.03–1.06) | <0.001 | 6,087/30,823 | 97,832/488,772 | 0.99 (0.96–1.02) | 0.485 | 666/3,153 | 9,495/47,652 | 1.07 (0.98–1.17) | 0.145 |
| Genitourinary | 5,754/25,730 | 286,187/1,433,975 | 1.15 (1.11–1.18) | <0.001 | 1,891/8,258 | 102,028/511,337 | 1.15 (1.09–1.21) | <0.001 | 163/807 | 9,998/49,998 | 1.02 (0.86–1.21) | 0.794 |
| Skin | 4,345/19,069 | 287,596/1,440,636 | 1.15 (1.12–1.19) | <0.001 | 1,450/7,232 | 102,469/512,363 | 1.01 (0.95–1.07) | 0.813 | 181/847 | 9,980/49,958 | 1.06 (0.90–1.25) | 0.480 |

AD, Alzheimer's disease; ALS, amyotrophic lateral sclerosis; CI, confidence interval; CNS, the central nervous system; OR, odds ratio; PD, Parkinson's disease.
Conditional on matching factors (sex and year of birth) and further adjusted for area of residence, educational attainment, family history of the disease, and history of comorbidity. Infections diagnosed during 5 years before the index date were excluded to alleviate the potential influence of reverse causation due to diagnostic delay and surveillance bias.

PD, $P < 0.001$), and were stronger during 1970 to 1986 (OR = 1.32, 95% CI: 1.27 to 1.36 for AD, $P < 0.001$; OR = 1.14, 95% CI: 1.08 to 1.20 for PD, $P < 0.001$), and among individuals born after 1960 (OR = 1.59, 95% CI: 1.52 to 1.65 for AD, $P < 0.001$; OR = 1.39, 95% CI: 1.24 to 1.56 for PD, $P < 0.001$) (S4 Table). Stronger associations were also observed for AD ascertained before 65 (e.g., for individuals with infections before 40 years, OR = 1.95, 95% CI: 1.91 to 2.00, $P < 0.001$), compared with AD ascertained after 65 (OR = 1.11, 95% CI: 1.05 to 1.18, $P < 0.001$) (S5 Table).

All these results were primarily attributable to AD and PD diagnosed before 60 years (Table 3), compared with those diagnosed at 60 years or older (S6 Table). For example, genitourinary infection was associated with the highest risk of AD before 60 years (OR = 2.75, 95% CI: 2.54 to 2.98, $P < 0.001$), whereas CNS infection was associated the highest risk of PD before 60 years (OR = 1.39, 95% CI: 1.15 to 1.69, $P < 0.001$). Gastrointestinal infection was also associated with a higher risk of AD (OR = 1.35, 95% CI: 1.28 to 1.41, $P < 0.001$) and PD (OR = 1.18, 95% CI: 1.06 to 1.32, $P < 0.001$) diagnosed before 60 years.

**Table 3. Association between hospital-treated infection and risk of neurodegenerative disease diagnosed before 60 years.**

| Group | AD | | PD | | ALS | |
|---|---|---|---|---|---|---|
| | OR (95% CI) | P | OR (95% CI) | P | OR (95% CI) | P |
| Analyses by characteristics of infections | | | | | | |
| Infection type | | | | | | |
| Bacterial | 2.08 (2.02–2.15) | <0.001 | 1.30 (1.21–1.39) | <0.001 | 1.09 (0.93–1.28) | 0.275 |
| Viral | 1.87 (1.79–1.94) | <0.001 | 1.36 (1.24–1.49) | <0.001 | 1.02 (0.84–1.24) | 0.816 |
| Others | 1.85 (1.73–1.98) | <0.001 | 1.33 (1.16–1.53) | <0.001 | 0.97 (0.71–1.33) | 0.844 |
| Infection site | | | | | | |
| CNS | 1.63 (1.49–1.78) | <0.001 | 1.39 (1.15–1.69) | 0.001 | 1.13 (0.78–1.64) | 0.516 |
| Gastrointestinal | 1.35 (1.28–1.41) | <0.001 | 1.18 (1.06–1.32) | 0.002 | 0.92 (0.72–1.17) | 0.494 |
| Respiratory | 1.63 (1.57–1.70) | <0.001 | 1.33 (1.22–1.46) | <0.001 | 1.12 (0.93–1.35) | 0.217 |
| Genitourinary | 2.75 (2.54–2.98) | <0.001 | 1.26 (1.04–1.52) | 0.016 | 1.29 (0.86–1.92) | 0.213 |
| Skin | 1.89 (1.77–2.02) | <0.001 | 1.16 (0.99–1.36) | 0.071 | 1.50 (1.12–2.02) | 0.007 |
| Age at infection | | | | | | |
| <40 years | 1.95 (1.90–2.00) | <0.001 | 1.36 (1.28–1.45) | <0.001 | 0.98 (0.84–1.13) | 0.731 |
| 40–59.9 years | 1.72 (1.62–1.82) | <0.001 | 1.08 (0.99–1.18) | 0.093 | 1.03 (0.84–1.25) | 0.797 |
| Age and frequency of infection | | | | | | |
| <40 years | | | | | | |
| 0 | Ref. | | Ref. | | Ref. | |
| 1 | 1.73 (1.69–1.78) | <0.001 | 1.27 (1.18–1.37) | <0.001 | 0.96 (0.81–1.13) | 0.595 |
| ≥2 | 2.70 (2.59–2.82) | <0.001 | 1.63 (1.47–1.82) | <0.001 | 1.02 (0.81–1.28) | 0.873 |
| 40–59.9 years | | | | | | |
| 0 | Ref. | | Ref. | | Ref. | |
| 1 | 1.64 (1.53–1.76) | <0.001 | 1.02 (0.92–1.14) | 0.692 | 1.13 (0.90–1.42) | 0.280 |
| ≥ 2 | 1.97 (1.76–2.20) | <0.001 | 1.21 (1.05–1.41) | 0.011 | 0.80 (0.55–1.15) | 0.229 |
| Stratified analyses by sex, calendar period, and birth cohort | | | | | | |
| Sex | | | | | | |
| Male | 1.76 (1.70–1.81) | <0.001 | 1.17 (1.08–1.27) | <0.001 | 1.07 (0.90–1.26) | 0.468 |
| Female | 2.18 (2.11–2.26) | <0.001 | 1.42 (1.31–1.54) | <0.001 | 0.98 (0.81–1.19) | 0.839 |
| Calendar period at diagnosis | | | | | | |
| 1970–1986 | 2.39 (2.26–2.52) | <0.001 | 1.42 (1.25–1.62) | <0.001 | 1.22 (0.80–1.86) | 0.360 |
| 1987–2000 | 1.88 (1.83–1.93) | <0.001 | 1.59 (1.45–1.75) | <0.001 | 0.80 (0.59–1.10) | 0.170 |
| 2001–2016 | 1.35 (1.20–1.52) | <0.001 | 1.04 (0.95–1.13) | 0.386 | 1.06 (0.91–1.23) | 0.458 |
| Year of birth | | | | | | |
| 1900–1919 | 2.08 (1.35–3.19) | 0.001 | 2.59 (1.49–4.48) | 0.001 | 2.30 (0.69–7.68) | 0.174 |
| 1920–1939 | 1.87 (1.75–1.99) | <0.001 | 1.30 (1.14–1.48) | <0.001 | 0.88 (0.60–1.31) | 0.539 |
| 1940–1959 | 2.15 (2.08–2.23) | <0.001 | 1.14 (1.05–1.23) | 0.001 | 0.97 (0.81–1.15) | 0.694 |
| ≥ 1960 | 1.59 (1.52–1.65) | <0.001 | 1.39 (1.24–1.56) | <0.001 | 1.11 (0.88–1.40) | 0.366 |

AD, Alzheimer's disease; ALS, amyotrophic lateral sclerosis; CI, confidence interval; CNS, the central nervous system; OR, odds ratio; PD, Parkinson's disease.
Conditional on matching factors (age and sex) and further adjusted for area of residence, educational attainment, family history of the disease, and history of comorbidity. Infections diagnosed during 5 years before the index date were excluded to alleviate the potential influence of reverse causation due to diagnostic delay.

## Secondary analyses

Compared to the main analyses, similar results were observed when restricting the analyses to individuals without a family history of the disease or when defining neurodegenerative diseases through at least 2 hospital visits concerning the same disease (S7 Table). We observed similar results in the model without adjusting for education and when restricting the analysis

to individuals with complete data on education, consistent results were also noted between the models with or without adjustment for education (S7 Table). Similar results were also observed after excluding individuals with multiple neurodegenerative diseases (S7 Table). Finally, after excluding infections experienced during 10 years before index date, similar associations were noted between any and specific hospital-treated infections and risk of AD and PD (S8 Table). For example, for any infection, the OR of AD and PD was 1.18 (95% CI: 1.16 to 1.19, $P < 0.001$) and 1.04 (95% CI: 1.01 to 1.06, $P = 0.002$), respectively. A dose–response relationship was still observed for frequency of infections after using a 10-year lag time ($P_{for\ trend} < 0.001$). The OR of AD and PD was 2.22 (95% CI: 2.12 to 2.33, $P < 0.001$) and 1.26 (95% CI: 1.15 to 1.38, $P < 0.001$) in relation to $\geq 2$ events of infection before age 40 (S9 Table).

## Discussion

In this nationwide study, we found that hospital-treated infections 5 years or more prior to neurodegenerative disease diagnosis were associated with an increased risk of AD and PD, specifically among cases diagnosed before 60 years, but not ALS. The associations for AD and PD were observed across infection types and sites but were stronger for infections—especially repeated infections—in early- or mid-life.

Though causality cannot be inferred from the study, the similar results for different infection types and sites might suggest that the potential underlying mechanisms of the observed associations are not specific to certain pathogens and raise the possibility that systemic inflammation might play a role in brain health, an idea that is supported by evidence from a previous study reporting a relationship between hospital-treated infection and vascular dementia and AD [17]. Although unexpected, our subsequent findings that hospital-treated infections were more strongly associated with risk of AD and PD before 60 years, compared to later, and that individuals with repeated infections in early- and mid-life had the greatest risk increment of AD and PD, are new and potentially important. We hypothesize that infectious events may be a trigger or amplifier of a preexisting disease process, leading to clinical onset of neurodegenerative disease at a relatively early age among individuals with disease predisposition [35–37]. Active monitoring and prevention of severe infections may therefore help to prevent or delay disease onset among high-risk individuals.

The association between infections and an increased risk of AD has support from previous studies. Evidence has suggested a link between herpes virus infection, including herpes simplex virus type 1 and varicella zoster virus, and AD and dementia [11], although a review showed that available findings were inconsistent with overall low-quality evidence [16]. Cognitive impairment has also been shown among individuals with neurological varicella zoster virus infection or Coronavirus Disease 2019 (COVID-19) [38,39]. However, a reduced long-term risk of dementia was observed among individuals with herpes infection treated with antiviral medications [40]. Another study reported a dose–response relationship between number of infectious diseases and dementia risk [17]. This study, however, did not explore the effect of age at infection and whether a dose–response relationship existed still for AD after considering the influence of potential reverse causation due to the preclinical stage and diagnostic delay of AD. Two studies used the same datasets from the UK reported an increased risk of dementia [18] or AD [20] among individuals with infectious diseases, without, however, reporting a clear dose–response relationship.

Similarly, a link between infections and PD has also been proposed. Previous studies have suggested a role of influenza [12] and hepatitis C virus [13] on the risk of PD, whereas antiviral treatment of hepatitis C virus infection might mitigate such risk [41]. Occurrence of parkinsonism after influenza and worsening of parkinsonian syndromes following COVID-19 have

also been described [42]. *H. pylori* infection was shown to be associated with an increased risk of PD [14,43], whereas eradication of this bacteria was shown to improve PD symptoms [44,45]. A recent study from Denmark reported an increased risk of PD with similar magnitude as the present study more than 10 years after exposure to infections; however, the result was not statistically significant (OR = 1.04, 95% CI: 0.98 to 1.10) [12]. These findings, together with the present study, support potential involvement of infection in the etiopathogenesis of AD and PD, although efforts are still needed to explain contradicting findings between studies [7,11,16].

In the present study, although positive associations were observed for AD and PD diagnosed before 60 years and for infections in early- and mid-life throughout the study period, the magnitude of the associations decreased over calendar period. There are likely multiple explanations. First, the treatment of infections might have changed during the study period, with presumably better treatment outcomes during later calendar periods than before. Similarly, the diagnosis of neurodegenerative diseases might have improved over calendar period, not only thanks to improvements in diagnostic tools, especially imaging and more recently also soluble biomarkers, but also due to improved societal awareness of these diseases and resources in neurology and geriatric care. Second, we included outpatient care data for the definition of hospital-treated infections since 2001, whereas only inpatient care data were available before then. Infections requiring specialized care, especially inpatient care, might have a different role on neurodegenerative diseases compared to infections not requiring such. Indeed, a previous study suggested a much weaker association between primary care–based infections, compared with hospital-treated infections, and the risk of dementia (hazard ratio = 1.02; 95% CI: 1.00 to 1.04 versus 1.99; 95% CI: 1.94 to 2.04) [18].

Our study did not support an association of hospital-treated infections with ALS risk. The null finding, however, does not rule out the possibility that milder infections not attended by specialist care might still be of importance. Previous studies have, for example, shown that infections might contribute to protein aggregation and mislocalization as well as glutamate excitotoxicity—known pathological processes of ALS [9]. Enterovirus RNA sequences have been found in the CNS of ALS patients [21,46], while disturbed gut microbiome composition [8] and increased use of antibiotics have also been observed among ALS patients [47].

Several mechanisms might explain the link between infection and neurodegenerative disease. As shown in animal research, infectious agents and their metabolites can promote aggregation of misfolded protein in neuron and its propagation from the periphery to the CNS, e.g., amyloid precursor protein and hyperphosphorylated tau protein in AD and alpha-synuclein in PD [1]. Infectious agents might also elicit inflammatory responses at infection site, resulting in production of pro-inflammatory cytokines and chemokines, which, like infectious agents, can cross the blood–brain barrier, enter the CNS, and elicit neuroinflammation through activating microglia and astrocytes [9]. Some neurotropic microbes, especially those with ineradicable infection, like herpesvirus, could not only trigger chronic neuroinflammation [48] but also directly infect neurons [7]. Prodromal nonmotor symptoms, including gastrointestinal symptoms and poor olfaction, have indeed been reported in PD patients years before onset of motor symptoms [49], whereas a lower PD risk has been suggested after vagotomy [50] and appendectomy [51].

## Strengths and limitations

Unlike most previous studies, our study did not focus on specific infectious diseases (e.g., influenza, pneumonia, or viral hepatitis), but instead studied all infections requiring hospital treatment. To our knowledge, it is the first to date to assess the associations of hospital-treated

infections—by type, site, age, and frequency—with risks of the 3 most common neurodegenerative diseases in the same population. Strengths of the study include the nationwide study design, large sample size, representativeness of patients with neurodegenerative diseases, and individually matched controls randomly selected from the general population. These strengths enabled us to perform informative subgroup analyses to demonstrate disease-specific results. Complete follow-up due to linkages to the national registers and the objective and prospective ascertainment of infection and neurodegenerative diseases minimized further selection and measurement biases commonly existent in observational studies.

There are also limitations. An infection might be secondary to undiagnosed neurodegenerative disease, especially during the years before diagnosis (reverse causation). Moreover, individuals with an ongoing infection might have a higher probability of being investigated for neurodegenerative disease, compared with others (surveillance bias). To address these concerns, we excluded infections experienced during 5 years before diagnosis in the main analysis and those experienced during 10 years before diagnosis in a sensitivity analysis, which rendered similar results.

Misclassification of hospital-treated infections and neurodegenerative diseases is another concern. Due to incomplete coverage of inpatient care data before 1987 and lack of outpatient care data before 2001 in the National Patient Register, some individuals with hospital-treated infections might have been misclassified as not having infection. Further, our definition of infections did not include those attended by primary care or not attended by healthcare at all. As a result, the present findings should only be interpreted in the context of relatively severe infections requiring specialist care. Besides, although register-based definition of infections was shown to have a high specificity (>95%) [22], we had no laboratory data to confirm the infections, due to the register-based nature of the study. In addition, as the sensitivity of register-based definition is low for AD (32.5%) [23], although higher for PD (72%) [24] and ALS (>90%) [25], not all patients with neurodegenerative diseases were identified, which might have diluted the real associations toward null.

The patient population included in the study consisted of all patients with a newly diagnosed neurodegenerative disease during 1970 to 2016 in Sweden. As suggested in previous validation studies [23–25], the National Patient Register is a valid data source for epidemiological studies of neurodegenerative diseases. However, as all cases were identified through specialist care, there was likely a delay comparing the register-based date of diagnosis (especially for the ones identified through inpatient care) and the actual date of diagnosis. Also, although the studied neurodegenerative diseases should theoretically all be referred to specialized care by a neurologist or geriatrician, we cannot exclude the possibility that some patients might have only been attended in primary care and misclassified as not having the disease in the National Patient Register. In addition, as we had relatively young individuals in the study, there was a high proportion of neurodegenerative diseases ascertained before 60 years. For example, 26% of the AD individuals were ascertained before 60 years (median age at diagnosis: 41; 56.4% of them born after 1956), higher than expected. However, the proportion was 13.3% among individuals born before 1956.

Residual confounding is also a concern as we did not have complete information on all risk or protective factors for neurodegenerative diseases, including lifestyle factors (smoking and body mass index), medical factors (e.g., human leucocyte antigen type [52] and brain damage due to vascular, traumatic, and illicit drug use reasons [53]), and genetic factors (e.g., ethnicity or risk genes [54]). Moreover, due to varying incidence of infections and neurodegenerative diseases across countries [2,3,8] and relatively young participants in present study, generalization of our findings to other settings should be done with caution. Similarly, whether hospital-treated infections indeed do not influence the risk of AD and PD beyond 60 years of age needs

to be studied further. Finally, future studies are needed to better understand roles of specific pathogens, infectious duration, and treatments (e.g., antibiotics, known to affect microbial environment and lead to dysbiosis [7]) on the link between infections and neurodegenerative disease.

In conclusion, our study suggested that individuals with hospital-treated infections, especially in those occurring in early- and mid-life, had an increased risk of developing AD and PD, attributable to cases diagnosed before 60 years. Further studies are warranted to validate these findings, to elucidate underlying mechanisms, and to determine whether better control of hospital-treated infections could prevent or delay onset of neurodegenerative diseases, especially the ones with an onset relatively early in life.

## Supporting information

**S1 STROBE checklist. STROBE statement—checklist of items that should be included in reports of observational studies.**
(DOC)

**S1 Table. The Swedish revisions of International Classification of Diseases (ICD) codes for neurodegenerative diseases.**
(DOCX)

**S2 Table. The Swedish revisions of International Classification of Diseases (ICD) codes for hospital-treated infections.**
(DOCX)

**S3 Table. Associations between age at hospital-treated infection and the consequent risks of neurodegenerative diseases (5-year lag time).**
(DOCX)

**S4 Table. Association between any hospital-treated infection and risk of neurodegenerative disease, stratified analyses by sex, age at diagnosis, calendar period at diagnosis, and year of birth (5-year lag time).**
(DOCX)

**S5 Table. Associations between age at hospital-treated infection and the consequent risks of early-onset AD and late-onset AD (5-year lag time).**
(DOCX)

**S6 Table. Association between hospital-treated infection and risk of neurodegenerative disease diagnosed at 60 years or older.**
(DOCX)

**S7 Table. Sensitivity analyses for the association between any hospital-treated infection and risk of neurodegenerative disease (5-year lag time).**
(DOCX)

**S8 Table. Association between any hospital-treated infection and risk of neurodegenerative disease (10-year lag time).**
(DOCX)

**S9 Table. Associations between age at hospital-treated infection and the consequent risks of neurodegenerative diseases (10-year lag time).**
(DOCX)

**S1 Text. Analysis plan.**
(DOCX)

## Author Contributions

**Conceptualization:** Fang Fang.

**Data curation:** Weimin Ye.

**Formal analysis:** Jiangwei Sun, Fang Fang.

**Funding acquisition:** Jiangwei Sun, Fang Fang.

**Investigation:** Weimin Ye.

**Methodology:** Jiangwei Sun, Fang Fang.

**Project administration:** Weimin Ye.

**Resources:** Weimin Ye.

**Software:** Jiangwei Sun.

**Supervision:** Caroline Ingre, Ulrika Zagai, Fang Fang.

**Writing – original draft:** Jiangwei Sun, Jonas F. Ludvigsson, Caroline Ingre, Fredrik Piehl, Karin Wirdefeldt, Ulrika Zagai, Weimin Ye, Fang Fang.

**Writing – review & editing:** Jiangwei Sun, Jonas F. Ludvigsson, Caroline Ingre, Fredrik Piehl, Karin Wirdefeldt, Ulrika Zagai, Weimin Ye, Fang Fang.

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
