## [Editor Report · Decision Letter 0]

7 Apr 2022

Dear Dr Sun, 

Thank you for submitting your manuscript entitled "Hospital-treated infections in early and mid-life increase the risk of Alzheimer’s disease and Parkinson’s disease" for consideration by PLOS Medicine.

Your manuscript has now been evaluated by the PLOS Medicine editorial staff and I am writing to let you know that we would like to send your submission out for external assessment.

However, we first need you to complete your submission by providing the metadata that is required. To this end, please login to Editorial Manager where you will find the paper in the 'Submissions Needing Revisions' folder on your homepage. Please click 'Revise Submission' from the Action Links and complete all additional questions in the submission questionnaire.

Please re-submit your manuscript within two working days, i.e. by Apr 11 2022 11:59PM.

Once your full submission is complete, your paper will undergo a series of checks in preparation for external assessment. 

Kind regards,

Richard Turner, PhD

rturner@plos.org

---

## [Decision Letter · Decision Letter 1]

8 May 2022

Dear Dr. Sun,

Thank you very much for submitting your manuscript "Hospital-treated infections in early and mid-life increase the risk of Alzheimer’s disease and Parkinson’s disease" (PMEDICINE-D-22-01054R1) for consideration at PLOS Medicine. 

Your paper was evaluated by an academic editor with relevant expertise and sent to independent reviewers, including a statistical reviewer. The reviews are appended at the bottom of this email and any accompanying reviewer attachments can be seen via the link below:

[LINK]

In light of these reviews, we will not be able to accept the manuscript for publication in the journal in its current form, but we would like to invite you to submit a revised version that addresses the reviewers' and editors' comments fully. You will appreciate that we cannot make a decision about publication until we have seen the revised manuscript and your response, and we expect to seek re-review by one or more of the reviewers. 

We hope to receive your revised manuscript by May 30 2022 11:59PM. Please email us (plosmedicine@plos.org) if you have any questions or concerns.

Please let me know if you have any questions, and we look forward to receiving your revised manuscript. 

Sincerely,

Richard Turner, PhD

Senior editor, PLOS Medicine

rturner@plos.org

Please revisit the data statement (submission form) to detail the source(s) of data and provide contact information for readers interested in seeking access. 

Please adapt the title to better match journal style: we suggest "Hospital-treated infections in early- and mid-life and risk of Alzheimer’s disease, Parkinson’s disease and amyotrophic lateral sclerosis: A case-control study".

Please specify the country in the title. 

Please restructure the abstract to combine the "Methods" and "Findings" subsections. 

Please add a new final sentence to the combined subsection, which should begin "Study limitations include ..." or similar and should quote 2-3 of the study's main limitations. 

In the abstract and throughout the paper, please quote p values alongside 95% CI, where available. 

After the abstract, please include and new and accessible "author summary" section in non-identical prose. You may find it helpful to consult one or two recent research articles published in PLOS Medicine to get a sense of the preferred style. 

Early in the Methods section (main text), please state whether or not the study had a protocol or prespecified analysis plan, and if so attach the relevant document as a supplementary file. 

Please highlight analyses that were not prespecified. 

Throughout the text, please format reference call-outs to the following style: "... genetic causes [6,7]." (noting the absence of spaces within the square brackets). 

In the reference list, please convert all italics and boldface text into plain text. 

Where appropriate, please list 6 author names, followed by "et al.".

Noting reference 5 and others, please ensure that all references have full access details. 

Please include a completed checklist for the most appropriate reporting guideline, e.g., STROBE, as an attachment, labelled "S1_STROBE_Checklist" or similar and referred to as such in the Methods section (main text). 

In the checklist, please refer to individual items by section, e.g., "Methods" and paragraph number, not by line or page numbers as these generally change in the event of publication. 

Comments from the reviewers:

*** Reviewer #1 (statistical reviewer): 

This study aims to examine the risk of three most common neurodegenerative diseases in relation to previous inpatient or outpatient episodes of hospital-treated infections. 

Comments:

"Hospital-treated infections, especially in early and mid-life, were associated with an increased subsequent risk of AD and PD, especially of AD and among cases diagnosed at relatively young age. These findings suggest that infectious events may be a trigger or amplifier of a pre existing disease process, leading to clinical onset of neurodegenerative disease at a relatively early age."

Can the authors please revisit the conclusions, being mindful that the study and analytical design does not allow causality to be inferred? 

"All individuals born after 1900 in Sweden whose parents were also born in Sweden were eligible for this study (N=12,275,551). We followed them from 1970 until a diagnosis of neurodegenerative disease, emigration, death, or December 31st, 2016, whichever occurred first, through cross linkages to the National Patient Register (NPR) and the Causes of Death Register, using the unique personal identity number."

Can the authors please clarify if they did or did not capture cases of onset of neurodegenerative disease before 1970?

Can the authors please comment on whether there may have been a change in diagnosis process and/or accuracy over time that could be impacting on the study inferences? 

Similarly, whether changes in individuals' capacity to treat infections over time might have an affect on the study findings?

"The analysis included 291,941 AD cases (median age at diagnosis: 76.2 years; male: 36 46.6%), 103,919 PD cases (74.3; 55.1%), and 10,161 ALS cases (69.3; 56.8%)."

Can the authors please comment on whether the analysed samples can be considered to be representative of the wider populations of interest?

"Individuals with multiple neurodegenerative diseases contributed to the analyses of different diseases."

Did the authors consider conducting a sensitivity analysis excluding individuals with multiple neurodegenerative diseases within each analysis?

"Five controls per case, individually matched by sex and year of birth, were randomly selected from the study base using incidence density sampling method."

A rigorous matching methodology has been applied by the authors.

"We then applied conditional logistic regression to estimate odds ratio (OR) and 95% confidence 135 interval (CI), as an estimate of the association."

The authors have applied technically appropriate modelling methods within the context of this research.

"Due to the concern about diagnostic delay, we used a lag time of 5 years, namely infections experienced during 5 years before the index date were excluded, in the main analysis. "

Can the authors please clarify how individuals with multiple cases of infection (< and > 5 years before diagnosis) were accounted for in the study? i.e. was the infection excluded if within five years of diagnosis, or the individual?

The authors have undertaken a thorough and extensive array of secondary and sensitivity analyses, which help to demonstrate the robustness of the study findings.

The Results are presented accurately and the main study limitations are transparently discussed.

*** Reviewer #2: 

This is a case-control study nested within national Swedish registers examining the associations of hospital-treated infections with Alzheimer's disease, Parkinson's disease and amyotrophic lateral sclerosis. The study is interesting and timely and benefits from a very large dataset. I also commend the authors for 5- and 10-year lag analyses to reduce the risk of reverse causation and ascertainment bias. My suggestions to improve the study are as follows:

Major points:

Because this is an observational study, I would suggest using more cautious language in the title. For example, "associated with an increased risk" would be more appropriate than "increase the risk", which implies causality. The title would perhaps also be more informative if it mentioned ALS as well.

The methods mention the specificity of the register-based outcome event definitions. It would be fair to report their sensitivity as well.

Regarding Table 1: I would assume the characteristics were recorded at index date rather than baseline. Please clarify.

Please clarify when were each of the covariates recorded. I would say that appropriate times for recording them would be on index date or earlier but not later (to prevent the covariates from being affected by the outcome).

Missing information in education was included as a separate category for adjustments. This approach would be fine if missing data were rare, but in this case up to half of the participants had missing information on education, which may lead to confounded estimates.

Please clarify, how was the index date defined in the analysis of at least two outcome events. Was it that of the first or second event? Where are these findings reported?

In the Discussion, the authors write that assessing only hospital-treated infections underestimates true associations. I think overestimation is also possible as hospital-treated infections are obviously more severe than those not requiring hospital treatment. This thought is also supported by a previous study, in which the association of primary care-recorded infections with dementia was very weak (Muzambi et al. Lancet Healthy Longev 2021;2(7):e426-35).

The first sentence of the Discussion states that individuals with neurodegenerative diseases were more likely to experience hospital-treated infections. This is confusing, because only 2 of the 3 assessed neurodegenerative conditions were associated with infections.

Vascular, traumatic and alcohol-related brain damage are common aetiologies of early onset dementia (e.g., McMurtray et al. Dement Geriatr Cogn Disord. 2006;21(2):59-64). Thus, in the interpretation of the findings, it should be taken into account that the observed associations of early and middle age infections with dementia could at least in part be explained by confounding due to vascular risk factors, head traumas or alcohol or other substance problems. Could these factors also be taken into account in the analysis of the data?

Minor points:

Please clarify why were only those whose parents were born in Sweden included in the study.

The statistical analysis section states that ORs from nested case-control studies mimic relative risk estimates of the underlying cohort study. To my understanding, this depends on the sampling method used unless the outcome is rare.

Although most previous studies focused on specific infections, this is not true of all the previous studies cited in the manuscript, although the first sentence of the strengths and limitations section of the Discussion (p. 16) seems to imply so.

Discussion, p. 17, lines 317-319: I agree that the register-based approach does not capture all patients with neurodegenerative diseases. However, contrary to what the text implies, this cannot necessarily be deduced from the low positive predictive values, but is rather dependent on the sensitivity of the outcome ascertainment procedure. 

The manuscript would benefit from language checking.

*** Reviewer #3: 

Summary

This was a clearly-written paper that used a nested matched case control design in Swedish National Patient Register data to compare the likelihood of hospitalised infections occurring more than 5 years prior to diagnosis between 291,941 cases of AD, 103,919 of PD or 10,161 of ALS and up to 5 controls per case matched on age and sex. Prior hospital-treated infections were more common among AD cases (1.16, 95% C.I. 1.15-1.18) and PD cases (1.04, 95% C.I. 1.02-1.06) but not ALS cases (OR 0.97, 95% C.I. 0.92-1.03). Associations were stronger for younger patients, and those with multiple infection episodes. 

Originality/ importance

Whether infections are a causal risk factor for neurodegenerative diseases is an important question with scope for major health impact. Existing longitudinal studies of clinically symptomatic infections and dementia have tended to focus on limited infection types e.g. sepsis or pneumonia1,2. There have been two recent large cohort studies from the UK and Finland, which showed positive associations between broad groups of medically-diagnosed infections and incident dementia risk3,4. This study extends findings to a different setting, with longer follow up and investigates two other neurodegenerative conditions, ALS and PD, as well as Alzheimer's disease. 

Specific comments:

1. In the methods section on p6, high figures are given for the specificity of register-based definitions of the three outcomes. It would be helpful to include figures for other aspects of validity such as sensitivity and positive predictive value, especially as the potential for outcome misclassification is later suggested as a possible limitation.

2. The study time period begins in 1970. While this is a strength in terms of length of follow up, there have presumably been major changes in diagnostic and recording practices for exposures, outcomes and covariates over that time period. Creating time strata of 1970-1986 and 1987-2016 may not be sufficient to highlight differences in results in different time periods. Did you consider stratifying further? Although there was some discussion of data completeness over time, it would be useful to consider this issue further.

3. Exposure was hospitalised infections occurring at least five years prior to the index date. It was unclear how infections occurring between 5 and 0 years before index were treated, and whether this might lead to exposure misclassification

4. For the AD analysis, over half of participants had missing data on education. How was this dealt with in the analysis?

5. Models were adjusted for a relatively limited number of confounding factors, which might have led to residual confounding. While a lack of data on factors such as BMI, smoking, alcohol was mentioned as a limitation, other potential confounding factors such as ethnicity and socioeconomic status were also not included (the latter would be incompletely captured by education or area of residence). In addition, the rationale for the approach to clinical comorbidities was unclear: why create a Charlson comorbidity index rather than describe and adjust for individual comorbidities separately? 

6. Health-seeking behaviour is likely to affect recording of both exposure and outcome. Was it possible to describe, or adjust for, frequency of consultations or healthcare contacts in the different exposure groups?

Minor comments:

1. The title should fully reflect the study - currently ALS is not mentioned.

2. Table 1 gave the proportion of the study population with any comorbidities, but it would be better to present Charlson comorbidity index scores (if this was the analytical approach taken)

3. The English language should be carefully checked throughout.

References

1. Muzambi R et al. Common Bacterial Infections and Risk of Dementia or Cognitive Decline: A Systematic Review. J Alzheimers Dis. 2020;76(4):1609-1626

2. Chu CS et al. Bacterial pneumonia and subsequent dementia risk: A nationwide cohort study. Brain Behav Immun. 2022 Apr 4;103:12-18

3. Muzambi R et al. Assessment of common infections and incident dementia using UK primary and secondary care data: a historical cohort study. Lancet Healthy Longev. 2021 Jul;2(7):e426-e435

4. Sipilä PN et al. Hospital-treated infectious diseases and the risk of dementia: a large, multicohort, observational study with a replication cohort. Lancet Infect Dis. 2021 Nov;21(11):1557-1567

*** Reviewer #4: 

"Hospital-treated infections in early and mid-life increase the risk of Alzheimer's disease and Parkinson's disease" (PMEDICINE-D-22-01054R1)

The objective of this paper was to examine "the risk of three most common neurodegenerative diseases in relation to previous inpatient or outpatient episodes of hospital-treated infections." Swedish national data were used, with a follow-up from 1970 until 2016. The results show hospital-treated infections to be associated with a higher risk of Alzheimer's disease (AD) and Parkinson's Disease but not amyotrophic lateral sclerosis. 

I have a number of concerns, listed below.

1. The statement "Infection is one of the most discussed potential non-genetic risk factors for neurodegenerative disease" (line 69-70) needs further elaboration as infection does not figure on most compilations of risk factors for neurodegenerative diseases, AD in particular.

2. In their review of the current status of knowledge the authors state that the role of "diagnostic delay" (line 78) has not been addressed in previous research. It isn't clear what this means, do they mean delay in diagnosis of neurodegenerative diseases? If this is the case, I did not find any analyses on diagnostic delay.

3. A major concern with this paper is what it adds to the recent paper by Sipila et al. Lancet Infect Dis 2021 (reference 17 in the manuscript). The measures of exposure and outcome, and conclusion are much the same.

4. The use of a lifecourse approach is promoted in the introduction but the authors chose to use a case-control design, this is a pity. Time to event analyses would have suited the data better; furthermore, the choice of controls retrospectively is never straightforward. It would be certainly better to retain everyone in the analyses so that people who die over the follow-up can be censored so that mortality does not bias findings.

5. There is no mention of period or cohort effects in the results; it is possible that these play a role, particularly for the exposure of interest.

6. In the primary analysis (lines 186-189) it is not clear if the results for men and women were different as statistical tests for difference are not provided.

7. Figure 3 shows that the association of infection and AD was confined to young-onset AD (under 60 years). These results are intriguing as the majority of AD cases in the population are late onset AD. Early onset AD is primarily driven by genetic factors and it is surprising to have 26% of cases in this study to be early onset AD. Please clarify the age at AD onset in this group; the median age at onset of AD in the total study population was 76.2 years (page 163).

8. It would be better to stratify results reported in Figure 4 as a function of early- and late-onset AD (after 65 years). 

9. The first sentence of the discussion (line 230-231) would be better supported using time to event analyses and a wash-out period of 20 years.

***

[LINK]

---

## [Decision Letter · Decision Letter 2]

26 Jun 2022

Dear Dr. Sun,

Thank you very much for submitting your revised manuscript "Hospital-treated infections in early- and mid-life and risk of Alzheimer’s disease, Parkinson’s disease, and amyotrophic lateral sclerosis: a nationwide nested case-control study in Sweden" (PMEDICINE-D-22-01054R2) for consideration at PLOS Medicine. 

Your paper was discussed with our academic editor, and re-sent to its four independent reviewers. The reviews are appended at the bottom of this email and any accompanying reviewer attachments can be seen via the link below:

[LINK]

In light of these reviews, we will again be unable to accept the manuscript for publication in the journal in its current form, but we would like to invite you to submit a further revised version that addresses the reviewers' and editors' comments fully. You will recognize that we cannot make a decision about publication until we have seen the revised manuscript and your responses, and we expect to seek re-review by one or more of the reviewers. 

We hope to receive your revised manuscript by Jul 15 2022 11:59PM. Please email us (plosmedicine@plos.org) if you have any questions or concerns.

Please let me know if you have any questions, and we look forward to receiving your revised manuscript. 

Sincerely,

Richard Turner, PhD

Senior editor, PLOS Medicine

rturner@plos.org

We suggest addressing referee 4's comments by reporting some additional analyses, perhaps in the supplementary information. 

Please reformat the 'Author summary' so that the 3 subsections each consists of around 3 bulleted points, each comprising 1-2 short sentences. 

At line 91 and any other instances, please use the general style "60 years". 

"Methods" at line 147.

At the end of the main text, please remove the information on funding from the Acknowledgements (in the event of publication this will appear in the article metadata, via entries in the submission form, and it does not need to be repeated here). 

Please make "p<0.001" the smallest p value quoted throughout, unless there are specific statistical reasons to do otherwise. 

Throughout the text, please move reference call-outs prior to punctuation, e.g., "... genetic causes [6,7]." at line 115.

Noting references 4 & 5, for example, please abbreviate journal names consistently. 

Noting reference 44, please use the journal name abbreviation "PLoS ONE". 

Comments from the reviewers:

*** Reviewer #1: 

The authors have satisfactorily considered and responded to each comment in turn, adding to the analysis and amending the manuscript accordingly.

*** Reviewer #2: 

The authors have done good job addressing my concerns. My few remaining concerns are as follows:

As a response to my question about the time of the recording of the covariates, the authors wrote that "[A]ll covariates were measured on the index date or earlier". I still struggle to understand what this means. Were there covariates for which data were not available up to the index date? If so, for what time period were the data available and used for defining each of these covariates? Please clarify in the manuscript the time periods used for the recording of the covariates.

In my initial review, I had concerns about missing data for education. I agree with the authors that conducting sensitivity analyses is the best way to address this issue. In the revised manuscript, they have conducted a sensitivity analysis in which they dropped the adjustment for education in the whole sample. This is not wrong, but does not take into account potential confounding from education. Therefore, I think the authors should conduct an additional sensitivity analysis among those with data available for education. Within this group, they should compare an analysis with adjustment for education to an analysis without adjustment for education. I think this would show whether adjustment for education would have any material effect on the results. 

I also have a couple of additional points that do not affect the validity of the revised manuscript but would clarify reporting:

Please specify in the methods whether the chosen controls were free from all three assessed neurodegenerative diseases or just the single disease of concern in each analysis (or if some other definition was used).

p. 17, lines 306-309: The authors write that "Similar results were observed 2) when restricting the analyses to individuals without a family history of the disease, or 3) when defining neurodegenerative diseases through at least two hospital visits concerning the same disease." Probably I missed it, but I could not find these results. Please report them in the appendix. Please also specify whether you mean that the results were similar to those in the age-stratified analysis described on lines 303-306 on the same page or to those of the main analysis.

*** Reviewer #3: 

Overall, I am satisfied that the authors have adequately addressed my comments. The new time-stratified analysis is helpful. I also agree with the approach to health seeking behaviour, which could indeed be acting a mediator in this context.

*** Reviewer #4:

[see attachment]

*** 

[LINK]

---

## [Decision Letter · Decision Letter 3]

29 Jul 2022

Dear Dr. Sun,

Thank you very much for re-submitting your manuscript "Hospital-treated infections in early- and mid-life and risk of Alzheimer’s disease, Parkinson’s disease, and amyotrophic lateral sclerosis: a nationwide nested case-control study in Sweden" (PMEDICINE-D-22-01054R3) for review by PLOS Medicine.

I have discussed the paper with my colleagues and the academic editor and it was also seen again by 3 reviewers. 

Provided the remaining editorial and production issues are dealt with we are planning to accept the paper for publication in the journal.

[LINK]

We look forward to receiving the revised manuscript by Aug 05 2022 11:59PM.   

Sincerely,

Caitlin Moyer, Ph.D.

Associate Editor 

PLOS Medicine

plosmedicine.org

Requests from Editors:

From the academic editor: 

1.The association is primarily being driven by the early-onset cases (<60 y). Once these are excluded, the OR is close to 1.0. Moreover, Fig 4 shows that 2 or more infections after 60 were in fact protective for AD ((OR 0.91 (0.89-0.93)), so to speak. It is true that the authors should look at the whole group first, but when they try to understand this deeper, the finding of early-onset stands out. This should be highlighted, and it is intriguing and counter-intuitive. The authors appear to de-emphasise this finding in favour of the whole group, but the message from the analysis appears to be that the finding is true only for early-onset cases.

Other editorial points:

2. Response to reviewers: Please address the remaining comments of Reviewer 4. Specifically, please discuss and present a figure/table of results broken down by age of diagnosis (i.e. older or younger than 60 years of age) for AD/PD/ ALS (sex, age at infection, type of infection, calendar period). Please also address the remaining issues raised by Reviewer 2. 

3. Title: Please revise as follows and please capitalize the first word of the subtitle and please make this change in the manuscript submission system as well as the manuscript file. “Hospital-treated infections in early- and mid-life and risk of Alzheimer’s disease, Parkinson’s disease, and amyotrophic lateral sclerosis: A nationwide nested case-control study in Sweden”

4. Competing interests: Please revise to “independent of the present study” in your statement.

5. Abstract: Line 33: Please mention the variables that are adjusted for in the analyses.

6. Abstract: Line 36-37: Please revise to: “A hospital-treated infection 5 or more years earlier was associated with…”

7. Abstract: Line 38-39: For the sake of comparison, please also present the results for those diagnosed at age 60 years or older. “The associations were primarily due to AD and PD diagnosed before 60 years”

8. Abstract: Line 44: Please delete “however” from the sentence.

9. Abstract: Line 51: We suggest “especially of AD” is not needed here as the association was apparent for both AD and PD. 

10. Author summary: Line 89-90: We suggest revising to: “The underlying mechanisms for the link between infections and neurodegenerative disease may not be specific to certain pathogens or affected organs but possibly occur at the systemic level.”

11. Methods: Lines 164-168: Rather than referring to sensitivity/PPV as “varying” please comment on the relatively lower sensitivity/PPV in particular for AD and why this might be the case: “The register-based definitions of AD, PD, and ALS have been validated against gold-standard clinical workup, showing a high specificity but a varying sensitivity and positive predictive value for AD (99.7%, 32.5%, and 57%) [23], PD (>98%, 72%, and 71%) [24], and ALS (all >90%) [25]. Date of diagnosis was defined as the date of first hospital visit concerning the disease.

12. Methods: Lines 167-168: Please clarify if the date of diagnosis was the first date where AD/PD/ALS was recorded as the primary diagnosis in the register.

13. Methods: Lines 175-176: Thank you for including a copy of the pre-specified analysis plan as a supporting information file. In the analysis plan, you note that updates were made to the plan in response to peer review. Please make sure that any changes in the analysis-- including those made in response to peer review comments-- are identified as such in the Methods section of the paper, with rationale.

14. Results: Lines 275-277: Please reference the figure/table where these results are presented: “Positive associations for AD and PD were similarly observed for bacterial, viral, and other infections, as well as for CNS, gastrointestinal, and genitourinary infections.”

15. Results: Lines 290-291: “Stronger associations were also observed for AD ascertained before 65, compared with AD ascertained after 65” Please present the OR, 95% CI and p values in the text for this finding.

16. Results: Line 299-302: Please mention the table/figure where the analysis with number of infections is presented.

17. Results: Line 312-313: “Gastrointestinal infection was also associated with a higher risk of

313 AD (OR=1.35, 95%CI: 1.28-1.41, P < 0.001) and PD (OR=1.18, 95%CI: 1.06-1.32, P < 0.001).” Please clarify if this is among all individuals or those diagnosed before age 60 years.

18. Results: Lines 321-346: In the description of secondary analyses, we suggest removing the numbering from the paragraph (“1) A stronger association…2) when restricting the analyses…).

19. Results: Lines 325-339: Please include supporting information tables presenting the results of analyses restricted to those without a family history, neurodegenerative diseases defined by at least 2 hospital visits, from models not adjusted for education or restricted to those with complete education data, and when those with more than one neurodegenerative disease were excluded.

20. Results: As suggested by reviewer 4, please also present a table of results broken down by age of diagnosis (i.e. older or younger than 60 years of age) for AD/PD/ALS (sex, age at infection, type of infection, calendar period). 

21. Discussion: Line 349-350, Please revise to: “In this nationwide study, we found that hospital-treated infections 5 years ago or earlier were associated with an increased risk of AD and PD, specifically among cases diagnosed before 60 years, but not ALS.”

22. Discussion: Line 355-357: Please revise to: “Though causality cannot be inferred from the study, the similar results for different infection types and sites might suggest that the potential underlying mechanisms of the observed associations are not specific to certain pathogens, and raise the possibility that systemic inflammation might play a role in brain health, an idea that is supported by evidence from a previous study reporting a relationship between hospital treated infection and vascular dementia and AD [17].” or similar.

23. Discussion: Line 385-387: Please revise this statement, as the paper cited actually reports finding no significant association between risk of PD more than 10 years after infection: “A recent study from Denmark reported an increased risk of PD more than 10 years after exposure to infections (OR=1.04, 95%CI: 0.98-1.10) [12].”

24. Discussion: Line 392-393: Please clarify if this refers to calendar year of diagnosis: “...the magnitude of the associations decreased over time.” Please comment on how the attenuation of the association in more recent calendar years might arise from better treatment for infection or improved diagnosis for AD/PD.

25. Discussion: Line 458-459: Please provide a reference: “The low sensitivity of AD diagnosis was likely attributable to misdiagnosis of AD as other dementias.”

26. Discussion: Line 470-473: Please comment on how the relatively young study population might impact the generalizability and interpretation of the findings, in light of the fact that associations were seen only among those diagnosed younger than 60 years of age.

27. Discussion: Line 478: Please use “substance misuse” here.

28. Discussion: Line 485-487: Please revise to: “In conclusion, our study suggested that individuals with hospital-treated infections, especially in those occurring in early- and mid-life, had an increased risk of developing AD and PD, attributable to cases diagnosed at young age.”

29. References: The journal abbreviation for reference 44 should be PLoS One.

30. STROBE checklist: For item 22, please refer to the location as “Funding” or similar.

31. Figure 1: Please define all abbreviations (AD/PD/ALS) in the legend.

32. Figure 2, Figure 3, Figure 4: Please define all abbreviations in the legend. Please add numbers of cases/controls and those with and without infection, as well as p values to the figure. Please note the factors adjusted for in the legends. Please also provide the results of unadjusted analyses (these can be added to the supporting information if preferred).

33. Table 2: Please add p values to the table. In the legend, please define all abbreviations used (AD/PD/CNS).

34. S3 -S10 Table: Please include p values. Please define AD, PD, ALS in the legends. Please also provide the unadjusted comparisons as well as the adjusted comparisons where applicable.

Comments from Reviewers:

Reviewer #1: The authors have satisfactorily considered and responded to each comment in turn, adding to the analysis and amending the manuscript accordingly.

Reviewer #2: The authors have responded to all my concerns, but their response needs to be clarified. When reporting the results of the sensitivity analyses suggested by me, they say that (p. 19, lines 333-336) "when restricting the analysis to individuals with complete data on education, consistent results were also noted between the models with or without adjustment for education (OR=1.10, 95%CI: 1.09- 1.12 for AD, P < 0.001; OR=1.03, 95%CI: 1.01-1.05 for PD, P < 0.001; and OR=0.98, 95%CI: 0.92-1.05 for ALS, P = 0.603)." Here, to support their text, they should show two ORs for each of the diseases (AD, PD, and ALS), one for the model adjusted for education and the other for the model not adjusted for education.

While reading the paper, I also noticed another point about clarifying the Discussion: On p. 21 (lines 372-376) the authors refer to an earlier study saying that "Another study reported a dose-response relationship between number of infectious diseases and dementia risk [17]. This study however did not explore the effect of age at infection and whether a dose-response relationship existed still after considering the influence of potential reverse causation due to the pre-clinical stage and diagnostic delay of neurodegenerative diseases." In a previous version of the manuscript they specified that the previous study did not consider the influence of potential reverse causation when looking at the dose-response relationship between dementia and AD, which was true. Now that the mention of AD has been omitted, the statement is not any more accurate as there was such an analysis on all-cause dementia in the previous study.

Reviewer #4: Hospital-treated infections in early- and mid-life and risk of Alzheimer's disease, Parkinson's disease, and amyotrophic lateral sclerosis: a nationwide nested case-control study in Sweden" (PMEDICINE-D-22-01054R3) 

1. Apologies for repeating what I said earlier but the revised version of the manuscript re-mains problematic. The unstratified analyses make no sense because of the number of difference as a function of age at exposure, birth-year, age at onset of the neurodegenerative diseases, and sex. The results section of the abstract provides results that are incorrect, see below

"A hospital-treated infection 5 years ago or earlier was associated with an increased risk of AD (OR=1.16, 95%CI: 1.15-1.18, P < 0.001) and PD (OR=1.04, 95%CI: 1.02-1.06, P < 0.001)."

2. This is also the case in the following sections of the authors summary; these statements suggest that infection is a risk factor for AD. Early onset AD is different from AD and the distinc-tion is lost in these sentences.

What did the researchers find?

Individuals with repeated infections in early- and mid-life had the greatest risk increment of AD and PD.

What do these findings mean?

The underlying mechanisms for the link between infections and neurodegenerative disease

are likely not specific to certain pathogens or affected organs but occur at the systemic level.

Infectious events may be a trigger or amplifier of a pre-existing disease process, leading to

clinical onset of neurodegenerative disease at a relatively early age among individuals with

disease predisposition.

3. In the response to reviewers comments the authors state the following: 

"We decided to report the unstratified analyses as the main results, because our primary hypothesis

was that there was an association between infections and risk of any AD/PD/ALS and these

analyses were a priori determined as the primary analyses. Although we did also hypothesize that

there might be variation in the associations by age at onset of neurodegenerative disease, etc., this

was a secondary hypothesis."

If this is the case then the authors ought to state clearly that their primary hypothesis was not verified and subsequent analyses showed an association for early onset AD/PD.

4. Following on from the previous comment, I don't quite understand the analyses reported in Figure 4. Surely, this ought to be stratified by age at onset of AD/PD.

[LINK]

---

## [Editor Report · Decision Letter 4]

15 Aug 2022

Dear Dr Sun, 

On behalf of my colleagues and the Academic Editor, Perminder Singh Sachdev, I am pleased to inform you that we have agreed to publish your manuscript "Hospital-treated infections in early- and mid-life and risk of Alzheimer’s disease, Parkinson’s disease, and amyotrophic lateral sclerosis: A nationwide nested case-control study in Sweden" (PMEDICINE-D-22-01054R4) in PLOS Medicine.

Please also address the following editorial points:

-Results: Line 279-280: Please remove “CNS” from the list here, as no significant associations were found between CNS infection and PD, in contrast with AD, according to Table 2.

-Discussion: Line 339: Please revise to: “...hospital-treated infections 5 years or more prior to diagnosis were associated with increased risk…” or similar.

PRESS

Sincerely, 

Caitlin Moyer, Ph.D. 

Associate Editor 

PLOS Medicine